# High nutrient loads amplify carbon cycling across California and New York coastal wetlands but with ambiguous effects on marsh integrity and sustainability

**Elizabeth Burke Watson**[1]*, **Farzana I. Rahman**[1¤], **Andrea Woolfolk**[2], **Robert Meyer**[2], **Nicole Maher**[3], **Cathleen Wigand**[4], **Andrew B. Gray**[5]

**1** Department of Biodiversity, Earth and Environmental Sciences and The Academy of Natural Sciences, Drexel University, Philadelphia, PA, United States of America, **2** Elkhorn Slough National Estuarine Research Reserve, Watsonville, California, United States of America, **3** The Nature Conservancy in New York, Uplands Farm Sanctuary, Cold Spring Harbor, New York, United States of America, **4** Atlantic Coastal Environmental Sciences Division, United States Environmental Protection Agency, Narragansett, Rhode Island, United States of America, **5** Department of Environmental Sciences, University of California, Riverside, California, United States of America

¤ Current address: Department of Earth & Atmospheric Sciences, University of Nebraska, Lincoln, Nebraska, United States of America

* elizabeth.b.watson@gmail.com

**Data Availability Statement:** All data presented in this manuscript has been uploaded as supplemental information, with included metadata.

## Abstract

Eutrophic conditions in estuaries are a globally important stressor to coastal ecosystems and have been suggested as a driver of coastal salt marsh loss. Potential mechanisms in marshes include disturbance caused by macroalgae accumulations, enhanced soil sulfide levels linked to high labile carbon inputs, accelerated decomposition, and declines in belowground biomass that contribute to edge instability, erosion, and slumping. However, results of fertilization studies have been mixed, and it is unclear the extent to which local environmental conditions, such as soil composition and nutrient profiles, help shape the response of salt marshes to nutrient exposure. In this study, we characterized belowground productivity and decomposition, organic matter mineralization rates, soil respiration, microbial biomass, soil humification, carbon and nitrogen inventories, nitrogen isotope ratios, and porewater profiles at high and low marsh elevations across eight marshes in four estuaries in California and New York that have strong contrasts in nutrient inputs. The higher nutrient load marshes were characterized by faster carbon turnover, with higher belowground production and decomposition and greater carbon dioxide efflux than lower nutrient load marshes. These patterns were robust across marshes of the Atlantic and Pacific coasts that varied in plant species composition, soil flooding patterns, and soil texture. Although impacts of eutrophic conditions on carbon cycling appeared clear, it was ambiguous whether high nutrient loads are causing negative effects on long-term marsh sustainability in terms of studied metrics. While high nutrient exposure marshes had high rates of decomposition and soil respiration rates, high nutrient exposure was also associated with increased belowground production, and reduced levels of sulfides, which should lead to greater marsh sustainability. While this study does not resolve the extent to which nutrient loads are negatively

**Funding:** The research described in this article has been funded by the U.S. Environmental Protection Agency, but has not been subjected to Agency review. Therefore, it does not necessarily reflect the views of the Agency. The funders had no role in study design, data collection, and analysis, decision to publish, or preparation of the manuscript.

**Competing interests:** The authors have declared that no competing interests exist.

affecting these salt marshes, we do highlight functional differences between Atlantic and Pacific wetlands which may be useful for understanding coastal marsh health and integrity.

## Introduction

Nutrient pollution to coastal areas continues to be an issue of concern in both developed and developing countries as global population growth continues [1] and population centers shift toward coastal areas [2]. Population growth increases wastewater, urban stormwater, and agricultural discharge to coastal zones, which can lead to symptoms of nutrient pollution and eutrophication. These include overgrowth of phytoplankton and opportunistic macroalgae [3], water column anoxia [4], and hyperoxic/anoxic diurnal cycles [5]. In addition, the enhanced labile carbon availability fuels the paired process of sulfate reduction and organic matter mineralization that occurs in anoxic environments [6]. Decreased light availability associated with eutrophication often reduces the extent of ecologically valuable seagrass meadows, and may negatively impact commercial and recreational fisheries and shellfish resources [7].

Previous studies have described both positive and negative impacts of enhanced nutrient availability on coastal wetland stability [8–11]. This uncertainty is problematic for land managers faced with coastal wetland drowning [12–16] due to uncertainty regarding the potential role anthropogenic nutrient enrichment may play in this loss [17]. As coastal marshes are known to be nutrient limited, fertilization has been found to enhance growth, productivity [11] and shoot density [18], which boosts sediment trapping and accretion via enhanced baffling [19]. However, a landscape-scale fertilization study has reported erosion of marsh channel banks exposed to nutrient concentrations typical of eutrophic estuaries [9,20]. Other long term fertilization studies have reported reductions in belowground biomass [21], soil strength [8], and carbon accumulation [22,23], especially in long-term plot-scale fertilization studies. Although drivers are not clear, rapid wetland loss has been reported for several eutrophic estuaries [12,24,25].

Impacts of enhanced nutrient availability on coastal marshes may operate through multiple mechanisms. Except in locations with carbonate soils and phosphorus is bound by calcite, coastal wetland plant communities are generally nitrogen-limited [10,26,27]. Thus, nitrogen subsidies may enhance growth both above and belowground, increase shoot density, lengthen growing seasons, and moderate stressful conditions for plants. However, as nitrate acts as a terminal electron acceptor, enhanced nitrate delivery to wetlands may enhance heterotrophic decomposition in concert with nitrate reduction (to $N_2$ or $NH_4^+$) [28,29]. Enhanced decomposition is problematic for marsh survival because a reduction in belowground organic matter stocks can threaten resilience to sea level rise [30]. High nutrient levels have also been associated with increased soil sulfide levels in wetlands [16,31,32], suggesting that increased labile carbon inputs associated with eutrophication may fuel the reduction of sulfate to sulfide, which acts as a toxicant to wetland plants [33]. Labile carbon inputs may also 'prime' microbial communities, catalyzing microbial decomposition and further reducing soil strength [34]. At the landscape level, higher nutrient availability may support blooms of opportunistic macroalgae and increase deposition of algal wrack which may negatively impact plant growth [18,35,36]. There is a need to better elucidate the relationship between eutrophication and coastal marsh sustainability, so managers can better understand the impacts of and manage nutrient pollution.

Additionally, the greenhouse gas mitigation value of wetland carbon sequestration [37] calls for more definition around potential benefits of nutrient reductions for soil carbon stocks. While studies have consistently found that added nutrients stimulates marsh decomposition both by enhancing litter quality and fertilizing, or altering the composition of microbial communities [28,38], studies have come to different conclusions about how fertilization affects carbon balance. While Morris and Bradley [23] found that 12 years of fertilizer addition to an oligotrophic marsh led to depletion of soil carbon stocks, recent studies have found conflicting results [22,39,40]. This confusion may be related to the difficulty with inventorying soil organic matter (SOM) in coastal wetlands, where soil profiles are accretionary and soil carbon densities are diluted through the deposition of mineral sediment at rates that vary heterogeneously within and between marshes [41]. However, as it has been found that nutrient additions can accelerate terrestrial carbon loss from aquatic [42,43] and agricultural ecosystems [44], understanding how nutrient pollution shapes carbon storages in soils is of broad interest across eco- and agricultural systems with similarly broad implications for sustainable management.

Here we focus on assessing a novel suite of metrics to reveal whether two eutrophic estuaries in California and New York that are undergoing rapid marsh loss [12,24] are experiencing the negative effects of eutrophication. By focusing on marshes in California and New York, we overcome issues with nutrient gradients that coincide with other gradients (such as that in salinity and tidal range) [17,45] and sample marshes with different ecosystem dominant species (*Spartina alterniflora* in New York and *Salicornia pacifica* in California). At Jamaica Bay, which is located in New York City between Brooklyn and Queens, decades to centuries of treated and untreated sewage discharges have interacted with other anthropogenic stressors (e.g., dredging; filling in; land development) causing alteration of the salt marsh system geomorphology and habitat integrity. At Elkhorn Slough, CA, which is surrounded by agricultural lands with high fertilizer usage, assessments of water quality have revealed eutrophic to hyper-eutrophic conditions, and these interact with other factors such as groundwater overdraft and diking to affect marsh elevations and integrity. At both marshes, coastal wetlands are disappearing. In NY, marshes are experiencing edge loss and expansion of the tidal channel network, with loss especially concentrated in marsh islands [12,46]; at Elkhorn Slough marshes have been lost to diking, which led to subsidence and conversion to mudflat after dike failure, as well as symptoms of excessive inundation [24]. While both places have experienced an increased tidal range due to dredging and in the case of Elkhorn Slough, construction of a deep water port [47,48], as well as loss of sediment supplies [49,50], these marshes experience extremely poor water quality. If poor water quality is a negative stressor to coastal marshes, we would expect these locations to be affected.

To test the hypothesis that high nutrient loads are affecting marsh soil integrity, and to determine effects of high nutrient exposure on carbon cycling, we measured a suite of metrics designed to assess production and decomposition. Soil integrity—or the capacity of soil to hold together and resist erosion [51] plays an important role in coastal marsh resilience to sea level rise. As direct indicators of carbon cycling, we measured belowground production, decomposition, and soil respiration. As additional indicators of long-term impacts on soil structure and integrity related to potential negative impacts of eutrophic conditions to wetlands, we also measured microbial biomass, SOM, soil humification, and porewater sulfide concentrations (Fig 1). To better compare these marshes, we reported water column nutrient data, measured porewater nutrients, soil and plant stable isotope data to assess nutrient exposure, and determined soil and environmental characteristics such as sediment texture that are likely covariates. We evaluate the results of our study within the context of previous work that

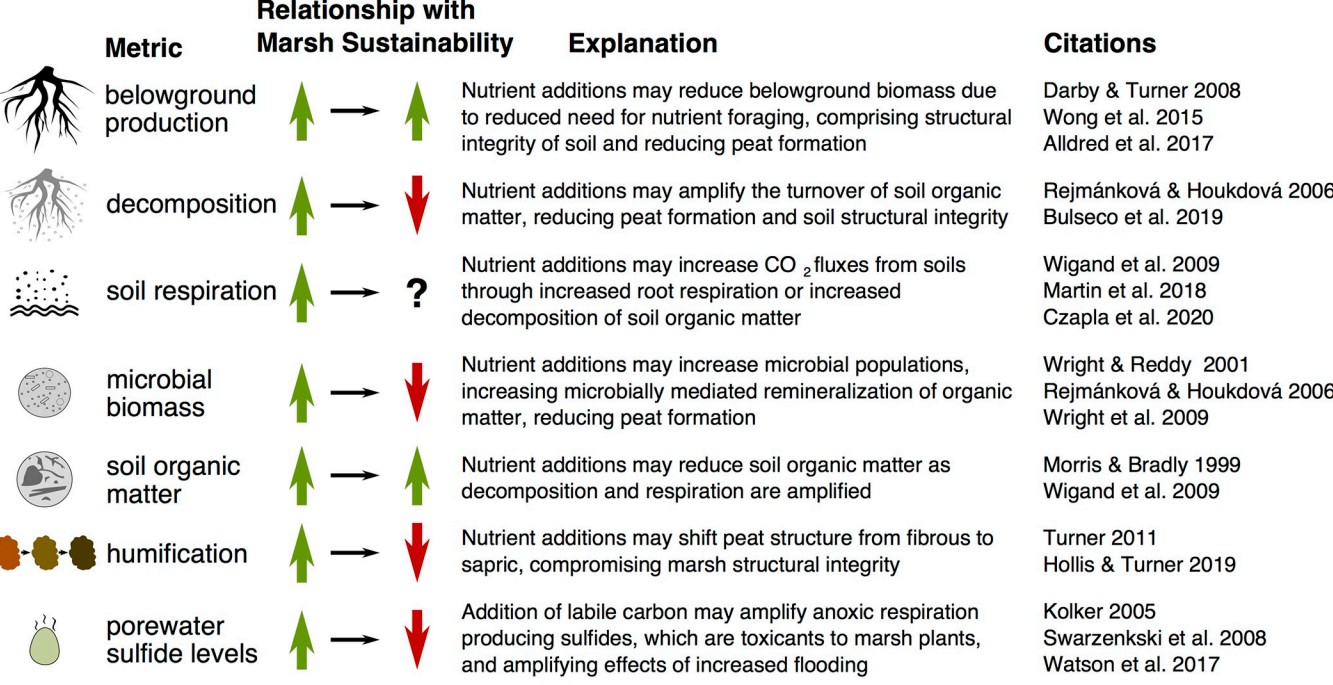

**Fig 1. Relationship between studied metrics, their impact on long-term marsh sustainability and integrity, an explanation of how nutrient additions or eutrophic conditions may alter such metrics, as well as references to peer reviewed publications that have associated metrics with high nutrient loads.** In the case of amplified soil respiration, such a metric may positively or negatively affect marsh sustainability, depending on whether this value is a response to enhanced root growth and respiration, or rather amplified decomposition of SOM. References: [16,22,23,28,31,38,52–60].

has presented opposing viewpoints on the impacts of eutrophic conditions on marshes and whether coastal marsh sustainability demands addressing nutrient pollution.

## Materials and methods

### Study sites

Eight coastal marshes were studied across four estuaries located in California and New York. Marshes were chosen that had similar vegetation, but possessed contrasting nutrient exposure. Also, because our previous work has suggested that flooding levels modulate effects of nutrient additions [18,32], both high and low marsh vegetation zones were sampled. Marshes studied included Jamaica Bay in New York City (Black Bank and Big Egg), two estuaries on Shelter Island, NY (Bass Creek and Mashomack Point Marshes), which have high and low nutrient exposure from cultural sources respectively. In California, we studied two sites in Morro Bay, CA (Chorro Creek Delta and Sweetwater Springs), and two sites at Elkhorn Slough (Monterey Bay, CA, Coyote Marsh and along the Old Salinas River Channel [OSR]) (Fig 2; Table 1). For the California estuaries, nutrient exposure varied within estuaries, with higher nutrient exposure at Sweetwater Springs and along the OSR, and lower nutrient exposure at the Chorro Creek Delta and at Coyote Marsh. Marsh loss has been observed at all sites, apart from Morro Bay, where data is lacking [12,24,32]. Field research permits were acquired from Gateway National Recreation Area, Morro Bay State Park, and the Moss Landing Harbor District. Permission for field sampling was acquired from the Elkhorn Slough National Estuarine Research Reserve, the Morro Coast Audubon Society, and the Mashomack Preserve.

Physical environment conditions varied between sites (Table 1). Mean salinity of porewater ranged from 26 psu in Jamaica Bay to 37 psu in Morro Bay. The California estuaries were

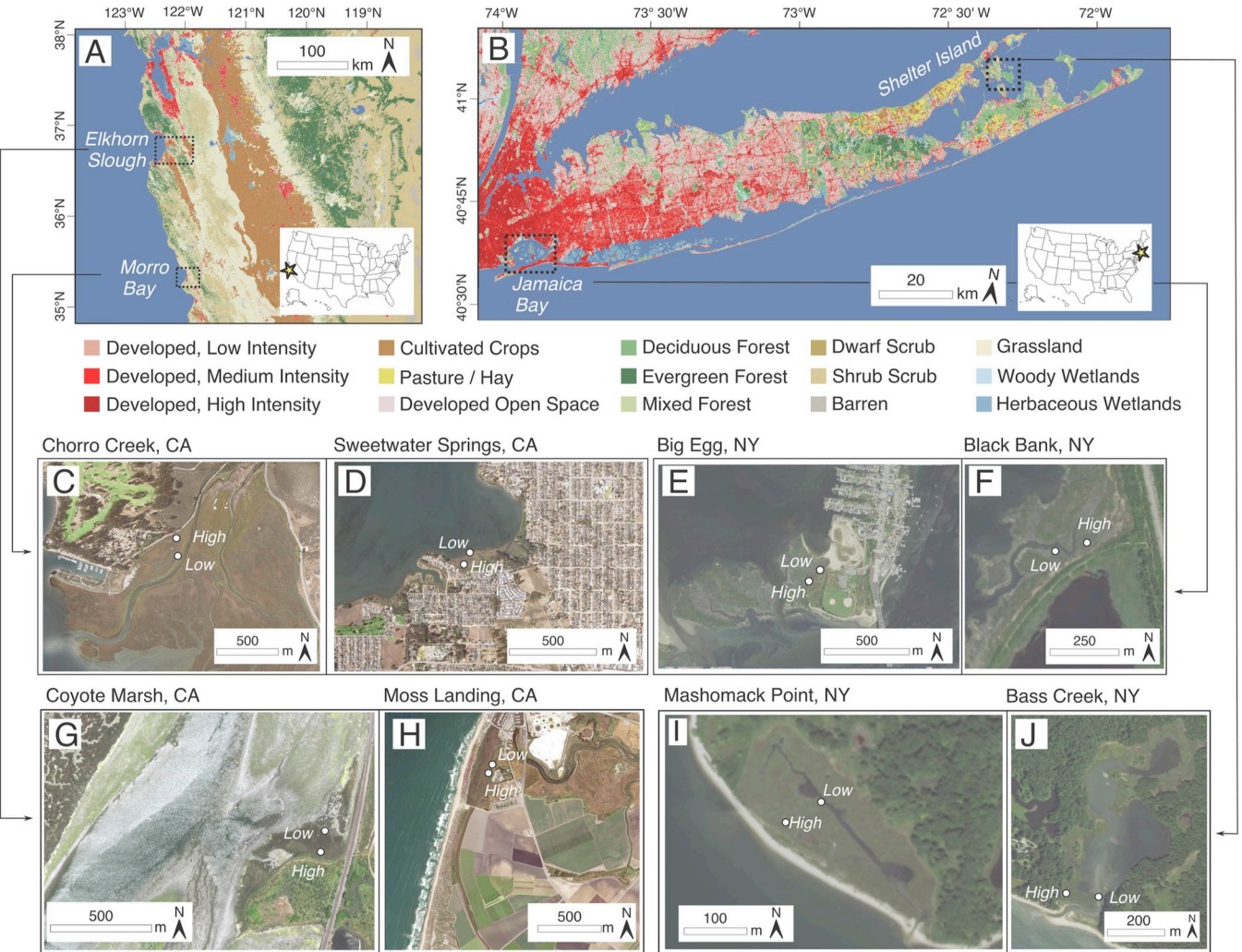

**Fig 2.** Location map of low and high marsh study sites in (A) California and (B) New York with land use, showing locations at (C) Chorro Creek and (D) Sweetwater Springs in Morro Bay, (E) Big Egg and (F) Black Bank at Jamaica Bay, (G) Coyote Marsh and (H) the Old Salinas River, at Elkhorn Slough, CA, and (I) Mashomack Point and (J) Bass Creek on, Shelter Island, NY [61].

dominated by *Salicornia pacifica* in both the high and low marsh; both lack the typical low salt-marsh species Pacific cordgrass (*Spartina foliosa*) found in many California estuaries. For the New York estuaries, the low marsh was dominated by tall-form *S. alterniflora* and the high marsh by short-form *S. alterniflora* and *Distichlis spicata*.

## Nutrient exposure and soil oxidation status

A range of properties were synthesized to represent nutrient exposure and soil oxidation status. Nutrient exposure was assessed through the compilation of publicly available water quality data (for dissolved nitrate, as ammonium data were not consistently available) [63–65], as well as through the measurement of surface soil and macrophyte stable nitrogen ratios. Macrophyte nitrogen stable isotope ratios were measured in a blended sample from leaf material of five plants (*Spartina alterniflora* at NY marshes; *Salicornia pacifica* from CA marshes) collected approximately 10m distance from each other at a mid-marsh location in August of 2013. Soil stable isotope ratios were measured on samples 0–3 cm in depth (also collected in August

**Table 1. Description of tidal marsh study sites generalized from average conditions across low and high marsh zones.** Tidal range and water column nitrate values are derived from [62–65]; while porewater salinity, soil texture, organic matter, soil and macrophyte $\delta^{15}$N were measured by this study.

| | Elkhorn Slough, CA | | Morro Bay, CA | | Jamaica Bay, NY | | Shelter Island, NY | |
|---|---|---|---|---|---|---|---|---|
| | **Coyote Marsh** | **OSR** | **Chorro Creek** | **Sweetwater** | **Black Bank** | **Big Egg** | **Mashomack** | **Bass Creek** |
| Location | 36˚49.79'N 121˚44.37'W | 36˚47.71'N 121˚47.29'W | 35˚20.75'N 120˚50.16'W | 35˚19.28'N 120˚50.81'W | 40˚37.33'N 73˚49.90'W | 40˚35.81'N 73˚49.57'W | 41˚1.69'N 72˚16.81'W | 41˚2.60' N 72˚17.48'W |
| Diurnal tidal range | 1.61 m | | 1.67 m | | 1.84 m | | 0.89 m | |
| Porewater salinity | 33 psu | | 37 psu | | 26 psu | | 27 psu | |
| Soil texture m m | silty clay clay = 20% silt = 71% sand = 9% | | silty clay clay = 22% silt = 70% sand = 8% | | silty sand clay = 6% silt = 25% sand = 67% | | sandy silt clay = 11% silt = 48% sand = 41% | |
| Soil organic matter | 34% | 16% | 11% | 28% | 16% | 14% | 35% | 27% |
| Macrophyte $\delta^{15}$N | 7.4‰ | 14.1‰ | 6.2‰ | 8.5‰ | 9.7‰ | 9.9‰ | 5.8‰ | 5.1‰ |
| Soil $\delta^{15}$N (0-3cm) | 6.5‰ | 11.9‰ | 4.8‰ | 10.1‰ | 8.7‰ | 9.3‰ | 2.0‰ | 2.3‰ |
| Water column nitrate | 15.4μM | 690μM | 2.0μM[a] | 579μM | 25.0μM | 21.9μM | 0.429μM | 0.178μM |
| Primary watershed land use | Row crop agriculture | | Open space | | High-density development | | Open space | |
| Watershed / size | Elkhorn Slough = 122 km$^2$ [c] | | 188 km$^2$ | | 368 km$^2$ | | 31 km$^2$ | |
| Dominant vegetation m | *Salicornia pacifica* | | *Salicornia pacifica* | | *Spartina alterniflora* *Distichlis spicata* | | *Spartina alterniflora* | |
| Yearly mean precipitation & temperature [b] | 54.7 cm 13.5˚C | | 42.5 cm 13.6˚C | | 110 cm 12.4˚C | | 126 cm 10.4˚C | |

[a] imputed from the relationship between natural log of mean water column nitrate values and soil surface $\delta^{15}$N for the seven other locations ($r^2$ = 0.92; p< 0.0001).

[b] Regional Climate Data Centers [66,67].

[c] Moro Cojo / Alisal = 463 km$^2$ intermittently receives water from Salinas (10,000+km$^2$).

2013), then processed for elemental abundance as stable isotope ratios as described above. Although stable isotopes are often used as tracers of wastewater, stable nitrogen isotope ratios are known to reflect overall nitrogen cycling, where the lighter $^{14}$N can be preferentially taken up and/or denitrified [68]. The values from these three assessments (macrophyte $\delta^{15}$N, soil $\delta^{15}$N, and water dissolved nitrate concentrations) were strongly inter-correlated ($r^2$ = 0.7–0.9). A principal components analysis was performed on the three variables (nitrate values were log-transformed prior to analysis), with the first principal component adopted as a nutrient exposure index (Fig 3). Soil oxidation status was a synthetic variable produced from the product of scaled particle size distribution (D50; described below) and marsh elevation values (relative to mean high water, MHW). Values were scaled from 0–1, with 0 representing the lowest elevation and finest sediment, with 1 representing the highest elevation and the coarsest sediment.

## Production and decomposition

**Belowground productivity.** Belowground production was measured using ingrowth bags [69] from August 2013 to August 2014. Mesh bags for root and rhizome ingrowth were constructed using fiberglass window screen (1.5 mm mesh) that measured 30 cm in length and 10.0 cm in diameter. Ingrowth bags were tubular in shape, and open at the top and sewn closed at the bottom. Bags were filled with native sand that was collected <1 km from deployment sites. Six ingrowth bags were deployed at each site, three in the low marsh and three in the high marsh (*n* = 48 in total). After collection, root and rhizome material was separated from sand, dried to constant weight, and weighed. Live and dead material were not separated. Weights were scaled to reflect productivity per 1 m$^2$.

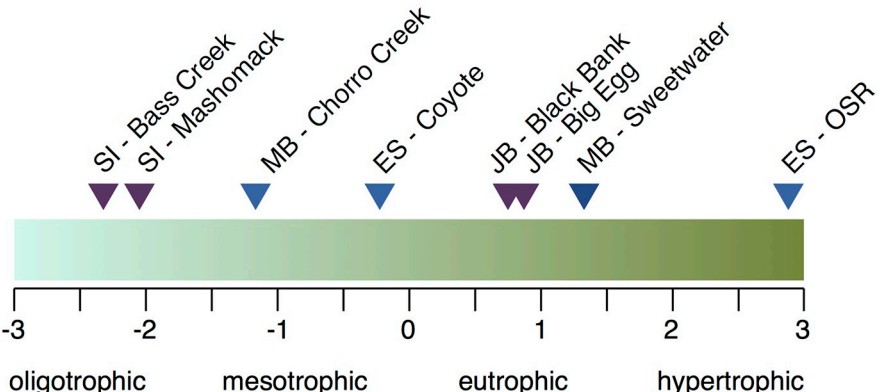

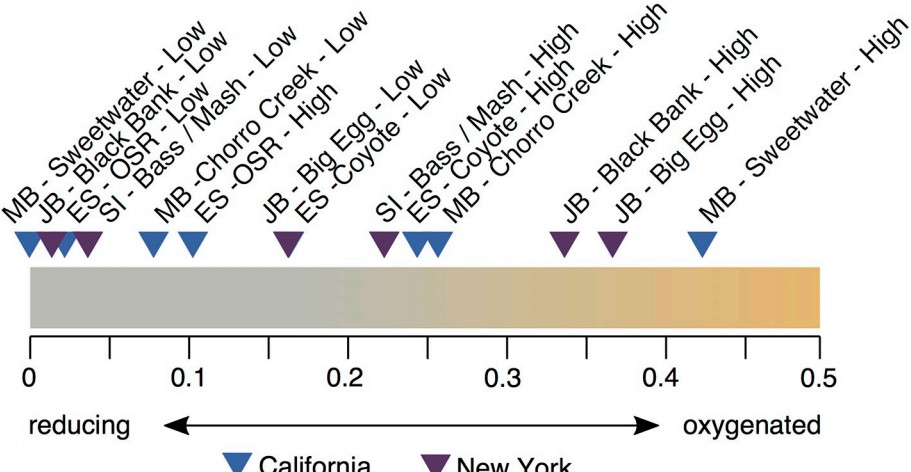

**Fig 3. Relative Nutrient Exposure for the eight study locations, at Elkhorn Slough (ES), Morro Bay (MB), Jamaica Bay (JB), and Shelter Island (SI), based on water quality nutrient and nitrogen stable isotope data.**

**Belowground decomposition.** Decomposition was measured for litter at the surface and at five sub-surface depths using a bag design with several separate pouches (0–2.5 cm; 5–7.5 cm; 10–12.5 cm; 15–17.5 cm; 20–22.5 cm). A total of 48 litter bags were deployed at each study site, 24 in the low marsh and 24 in the high marsh (total of 384 bags with six pouches each, for an overall $n$ = 2304). Half the bags were filled with dried (at 60°C) *S. alterniflora* leaf and belowground biomass material and half filled with dried *S. pacifica* leaf and belowground bio-mass material, as we wanted to test both vegetation types in both regions. Thus, we could test values for locally relevant vegetation but also compare consistently across regions using the same currency. Because litter decomposition rates are known to vary with litter quality [38,70], we held constant the site of litter collection: *S. alterniflora* material was collected from Galilee, Rhode Island (41.38° N, 71.49° W), and *S. pacifica* material was collected from Elkhorn Slough (36.81° N, 121.79° W). Litter bags were collected every two months over a 1.25-year period. Bag collections were terminated after 1.25 years as the amount of litter remaining in above-ground pouches was exhausted and belowground pouches began to increase in weight with

root ingrowth. Decomposition rates were calculated from the percentage of dry mass remaining using an exponential decay model [71,72]: $W_t/W_o = e^{-kt}$, where $W_t/W_o$ is the fraction of initial mass remaining at time $t$ (%), $t$ is the elapsed time (y) and $k$ is the decomposition constant ($y^{-1}$). The reported litter turnover time ($\tau$) was calculated as $1/k$, or the mean litter lifetime.

**Soil respiration.** Carbon dioxide efflux was measured in the low marsh at each study site during summertime peak respiration rates using a LiCor 8100 infrared gas analyzer outfitted with an opaque dome [30], with 10 cm PVC collars inserted in the soil approximately one hour before measurements were made. Incubations lasted three minutes, with observations of $CO_2$ concentration made every second. Soil respiration measurements were collected during summer as they have been observed to peak during summer due to high soil temperatures and availability of labile carbon exuded by roots [73,74]. Soil temperatures at 5 cm of depth were similar among sites ($23 \pm 4.0°C$) (mean ± standard deviation), and no correlation was observed between respiration rate and temperature ($r = 0.04$). Soil respiration was measured in the low marsh between clumps of marsh vegetation. Twelve locations were sampled per site, at least 10 m apart, for a total of 96 observations across the eight sites. Linear regression was used to calculate respiration rates based on change in $CO_2$ concentration over time.

**Soil humification.** Two soil cores 20–30 cm long from each site (one each from low and high marsh) were sectioned into 3-cm intervals for humification analysis. Humification refers to the abundance of humic substances in soil, which form as plant remains decay. Humification analysis is often used in analysis of peatland cores as an indicator of paleohydrology as low water tables allow for proliferation of aerobic micro-organisms which are better able to decompose lignocellulose than their anaerobic counterparts [75]. Soil core material was analyzed colorimetrically using the Blackford and Chambers method [76] with modifications by Borgmark [77]. Dried soil samples were placed into 50 ml plastic tubes and dissolved in 25 ml of 8% NaOH solution. The samples were boiled in a water-bath at 95°C for 1.5 hours. The samples were then vacuum-filtered and diluted based on the Borgmark [77] rate of 12.5 ml of sample to 100 ml of deionized water. Each sample was measured three times on a Shimadzu UV-1610 spectrophotometer at 540 nm. Values were detrended relative to organic content values to account for sediments that varied widely in organic content [78].

## Biogeochemical analyses

**Soil C, N, and δ$^{15}$N.** Two soil cores 20–30 cm long from each site (one each from low and high marsh) were sectioned into 3-cm intervals and analyzed for organic content and bulk density using loss on ignition [79]. Sediments were homogenized, dried, ground and analyzed for carbon, nitrogen, and stable isotopic ratios (δ$^{15}$N, δ$^{13}$C) using a Vario Cube elemental analyzer interfaced to an Isoprime 100 isotope ratio mass spectrometer (IRMS). Isotope ratios for carbon and nitrogen are reported in permille notation as: $\delta aX = \left(\frac{R_{sample}}{R_{standard}} - 1\right) \times 1000‰$ where $R$ is the abundance ratio of the less common ($a$) to more common isotope. The standard for N is atmospheric N gas; the standard for C is PeeDee Belemnite; by definition, standards have δ = 0. Mean difference between duplicates was 0.19% for C, 0.014% for N, and 0.35‰ for δ$^{15}$N.

**Microbial biomass.** Summer microbial biomass (C, P) was measured using the fumigation-extraction method at each litterbag deployment location following Brookes et al. [80,81] with modifications by White and Reddy [57]. Four soil samples (0–3 cm) were collected within one meter of litterbag deployment sites; paired samples were either fumigated or not fumigated and extracted with 25 mL of 0.5M $K_2SO_4$, (for C, N) or 0.5 M $NaHCO_3$ (for P). Fumigated samples were incubated for five days in evacuated dark desiccators with 100mL ethanol-

free chloroform. Samples were centrifuged, vacuum filtered, and analyzed for C using a Shimadzu TOC Analyzer, and for TP by analyzing 10 mL extracts for $PO_4^{3-}$ colorimetrically using an Alpkem Autoanalyzer. Extracts analyzed for microbial biomass values were calculated using extraction efficiency factors estimated by previous studies—0.37 for C [82], 0.4 for P [80]; and reported relative to dry weights.

**Porewater measurements.** Porewater was monitored seasonally at 16 locations across sites over one calendar year using passive-diffusion porewater samplers [83]. Samplers were constructed of 5 cm diameter PVC pipe screened at 15, 30, and 45 cm of depth. Inner casings held scintillation vials filled with deionized water capped with 45 mm Nitex screen aligned with screening depths, allowing the deionized water to equilibrate with porewater. After collection, porewater salinity and pH were measured. Porewater was sub-sampled, and preserved (1:1) with a 0.22% solution of zinc acetate ($Zn(O_2CCH_3)_2$) for hydrogen sulfide analysis. The remaining porewater was acidified and frozen for later nutrient analysis. Porewater samples were analyzed for hydrogen sulfide concentration colorimetrically using a Genesys 2 spectrophotometer [84,85], and for nitrate + nitrite, ammonium and phosphate using an Astoria Pacific A2 micro-segmented flow autoanalyzer (U.S. EPA methods 350.1, 353.2, and 365.2).

## Covariates

We conducted several analyses to quantify potential confounding environmental factors, including inundation, tidal range, and soil texture. Carbon mineralization rates are known to respond to oxygen availability, which covaries in wetlands with flooding and soil texture [86].

Flooding affects oxygen availability and therefore carbon mineralization in wetland soils, as water fills void spaces in soil, such that oxygen must diffuse through water at a rate that is significantly slower (8400 times) than through air. This results in a depletion of oxygen where metabolism consumes oxygen more quickly than it can be replenished [87]. Particle size distribution affects carbon mineralization directly as labile carbon can be occluded by aggregates or adsorbed to clays, which restrict the accessibility by microbes [88], and indirectly through the dependence of soil hydraulic properties on soil texture [89], which can influence saturation and dissolved oxygen regimes.

To quantify soil oxidation status at each site, elevation was measured using static post processed kinetic (PPK) surveys conducted with a Trimble 4700 or 5700 survey grade GPS receiver post-processed using OPUS. Elevations were converted to tidal heights (the height relative to mean high water, MHW) using VDATUM [62]. VDATUM was also used to estimate tidal range, as the elevation between mean higher high water (MHHW) and mean lower low water (MLLW), or the great diurnal range. Soils were sub-sampled at 3-cm intervals and analyzed for particle size distribution using a LS 13–320 Beckman Coulter laser granulometer with polarized intensity differential scattering (diameter, D, from 0.04 to 2000 µm in 117 bins) after pretreatment with heated hydrogen peroxide to remove coarse organic matter present in the soil matrix [90]. Soil texture values (%) were aggregated from particle size distributions on the basis of the following thresholds: clay ($D \leq 2$ µm), silt ($2$ µm $< D \leq 63$ µm), sand ($63$ µm $< D \leq 2000$ µm).

## Data analysis

Linear models were used to examine the effect of nutrient exposure and soil oxidation status on variables related to biomass production and decomposition. In these models, response variables were: belowground production, litter turnover time, microbial biomass, carbon dioxide efflux, and soil transmissivity. Predictors included continuous variables: nutrient exposure index (the first principal component of three related nutrient variables as described above)

and soil oxidation status, and categorical variables of location (CA vs. NY) and marsh zone (low vs. high). Porewater sulfide values were log-transformed prior to analysis using the form $(\log_{10}(Y + 1 - \min(Y))$. A correlation matrix was used to examine the interdependence of variables. All data analyses were conducted in R (version 3.6.1) [91] using R Studio (version 1.1.463) and package corrplot [92]. Based on rates of belowground production, and turnover rates calculated for the decomposition bags, soil carbon inventories were estimated using a time step model, adapting previous approaches [47,93], where a SOM inventory was estimated using productivity and an exponential decay function: $SOM = \sum_{i=0}^{t} Pe^{-k}$, where $i$ represents time step, $P$ is production and $k$ is the decomposition coefficient (both measured empirically) over a 20 year period ($t$).

## Results

### Production and decomposition

We found that nutrient exposure generally increased belowground production and increased decomposition rates (Fig 4; Table 2). In California, belowground production was 70% higher in the marsh with the highest nutrient exposure vs. the lowest nutrient exposure. In New York, belowground production was 140% greater in the marshes with higher nutrient exposure vs. marshes with lower nutrient exposure. Although nutrient exposure enhanced productivity, it also enhanced decomposition. The organic matter turnover time for buried litter bags was 3.7 years for the marsh with the lowest nutrient exposure, and 1.6 years from the marsh with the highest nutrient exposure. For leaf litter above the soil surface (including both species), the organic matter turnover time was 0.62 years in marsh with the lowest nutrient exposure vs 0.33 years in marshes with the highest nutrient exposure. High nutrient exposure was associated with soil-decomposition rates that were 130% greater, and for aboveground biomass decomposition rates that were 87% greater than found under lower nutrient availability. Based on litterbag analyses, we additionally found that *Spartina* aboveground biomass decomposed significantly more rapidly than *Salicornia* aboveground biomass, but that *Spartina* belowground biomass decomposed more slowly than *Salicornia* belowground biomass. This trend is unsurprising due to the woody nature of *Salicornia* aboveground material, but finer roots and rhizomes of belowground biomass. Belowground decomposition rates were also significantly more rapid in California—by 38%—than rates in New York (Table 2).

Soil carbon dioxide efflux, which reflects a combination of soil root respiration and decomposition, followed trends apparent in the productivity and decomposition data. Carbon

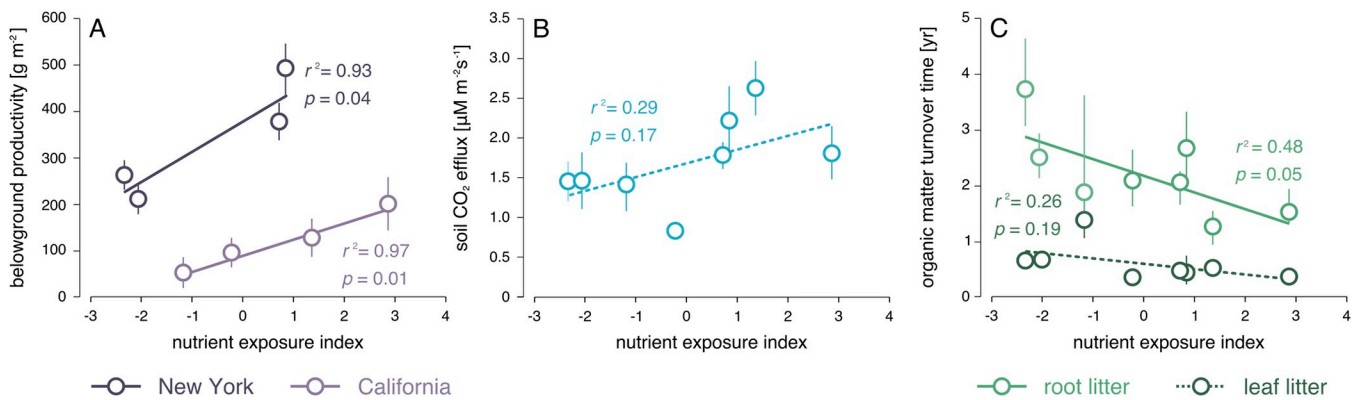

**Fig 4.** Relationship between nutrient exposure and (A) marsh belowground vegetation productivity; (B) soil $CO_2$ efflux, and (C) leaf and root turnover time.

**Table 2. Linear model results.**

| Measure | $r^2$ | Nutrient Exposure | | Marsh Zone (High/Low) | | Location (CA/NY) | | Soil oxygenation | | Significant Trends |
|---|---|---|---|---|---|---|---|---|---|---|
| | | t | p value | t | p value | t | p value | t | p value | |
| belowground production | 0.39 | **2.90** | **0.006** | -0.961 | 0.342 | **4.78** | **<0.001** | 1.35 | 0.19 | higher with higher nutrient exposure; higher in NY |
| decomposition—aboveground | 0.17 | **-2.32** | **0.034** | -0.639 | 0.528 | -1.21 | 0.233 | 0.431 | 0.669 | aboveground biomass decomposed faster under higher nutrients; *Spartina* aboveground biomass degraded 18% faster than *Salicornia* (t = -7.28, p<0.0001) |
| decomposition—belowground | 0.11 | -1.03 | 0.31 | -1.21 | 0.23 | **2.00** | **0.047** | 0.147 | 0.88 | plants decomposed faster in California; *Salicornia* belowground biomass decomposed 17% faster than *Spartina* (t = 12.48, p < 0.0001); slower decomposition with depth (t = 2.76, p = 0.0065) |
| $CO_2$ efflux | 0.11 | **3.35** | **0.006** | - | - | 1.27 | 0.21 | -1.40 | 0.16 | higher with higher nutrient exposure |
| soil humification | 0.35 | **-2.76** | **0.007** | **2.87** | **0.005** | **2.50** | **0.01** | -0.71 | 0.48 | soil was more decomposed under higher nutrients, at low elevations, and in California |
| microbial biomass | 0.26 | **-3.46** | **0.001** | -0.885 | 0.38 | 0.383 | 0.70 | 1.87 | 0.07 | higher with lower nutrient exposure |
| porewater sulfide | 0.40 | **-2.10** | **0.03** | **-4.07** | **<0.001** | **7.75** | **<0.001** | 0.24 | 0.81 | higher under low nutrient exposure, and at low elevations; higher in NY; higher at greater depths (t = 2.998; p = 0.03) |
| porewater DIN | 0.13 | **2.87** | **0.005** | 1.09 | 0.28 | 1.75 | 0.08 | **-2.05** | **0.043** | higher under higher nutrient exposure; lower at greater soil oxidation |
| porewater orthophosphate | 0.38 | **3.71** | **0.0004** | 1.21 | 0.23 | **2.34** | **0.023** | **-3.70** | **0.0004** | higher under higher nutrient exposure; higher in NY; lower at greater soil oxidation |

Bold values indicate statistical significance. Carbon dioxide efflux measures were only completed in the low marsh.

dioxide emission rates were 53% greater in the two marshes with the greatest nutrient exposure in comparison with the two marshes with the lowest nutrient exposure. Soil humification measures, which reflect how decomposed the SOM is, indicated that nutrient exposure resulted in more decomposed soils (Table 2). In addition, soils were more decomposed at lower elevations, and in California (Table 2), which is congruent with the observation that belowground decomposition rates were more rapid in California.

## Biogeochemical analyses

Sediment core stratigraphy showed that the majority of NY wetlands were underlain by sand or muddy sand, with a relatively thin (10–20 cm) veneer of peat at the marsh surface associated with finer particle sizes (Fig 5). Conversely, CA wetlands were composed of finer sediments, mud or sandy mud, with little, if any, variability in carbon concentration downcore. The low marsh was often associated with lower and more decomposed soil carbon (Fig 4; Table 2), although litter decomposition rates were not significantly higher in the low marsh (Table 2).

Microbial biomass was significantly greater at lower levels of nutrient exposure (Table 2), and was correlated with SOM, suggesting that substrate was more important to supporting microbial populations than the rate of organic matter cycling. Porewater sulfide concentrations were higher in the low marsh, greater at greater depths, and were greater in NY than CA marshes. There was a negative relationship with nutrient exposure, such that marshes exposed to lower nutrient concentrations had higher sulfide concentrations.

Porewater nutrient concentrations were related to overall nutrient exposure (Table 2), however, the relationship was stronger for orthophosphate than for dissolved inorganic nitrogen, presumably due to nitrogen assimilation by macrophytes. Nitrate values were greater in the CA marshes, suggesting more oxygenated soil conditions, and soil dissolved inorganic nitrogen (DIN) and orthophosphate concentrations varied with soil oxygen status, with higher values for porewater nutrients under more reducing conditions (Table 2). Iron oxides bind

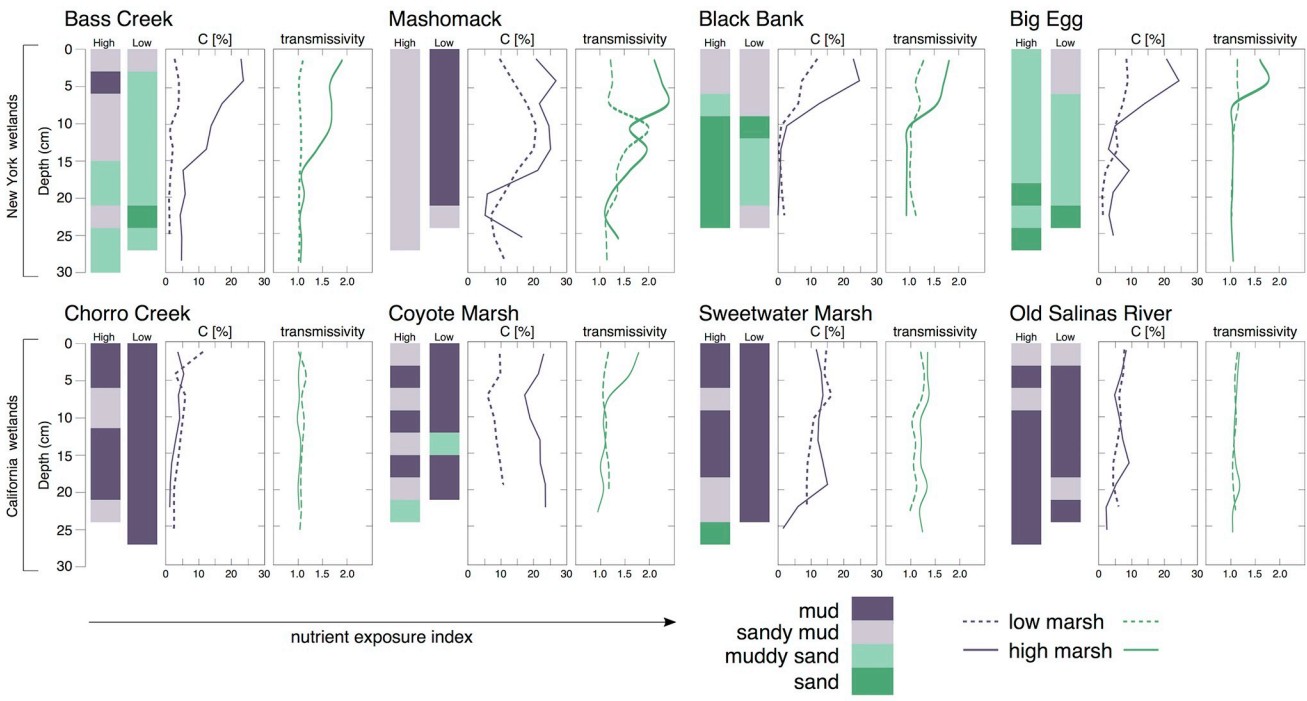

**Fig 5. Soil core profiles from study marshes, showing sediment type (indicated by color legend), percent carbon (C) and soil humification in the low and high marshes.** High transmissivity indicates less humified soils, and values have been corrected for organic content [78].

phosphorus, but release it under reducing conditions when iron oxides are reduced to ferrous iron [94]. DIN values were also higher under more reducing conditions (Table 2), possibly as plants growing under reducing conditions are less N-limited than plants growing more vigorously under more oxygenated conditions. Porewater orthophosphate concentrations were also found to be higher in NY than CA.

## Effects on soil carbon

Given that nutrient exposure was found to enhance both productivity and decomposition, how then does nutrient exposure affect soil carbon? For the sites we studied in New York, soil carbon and organic matter values were greater in the two marshes with the lower nutrient exposure (e.g., 31% organic vs. 15%). In California, marshes with very high and very low nutrient exposures had low values for SOM and soil carbon, with high values at moderate exposures. A comparison of SOM inventories measured through sediment analysis and estimated using measures of productivity and decomposition found that while there was a strong relationship between actual and predicted values for SOM inventories ($r^2 = 0.61$, $p < 0.01$), the predicted values were underestimates for Californian wetlands, and there was no apparent relationship with nutrient exposure. Variability in sediment texture is related to SOM (Fig 5).

## Discussion

### Effects of nutrient exposure

This study was motivated by observations of marsh drowning at Elkhorn Slough, California, and Jamaica Bay in NYC. Because these wetlands are exposed to high nutrient loads, we hypothesized that their marshes are experiencing symptoms of eutrophication, such as

enhanced rates of organic matter decomposition, low belowground production, reduced soil strength, and enhanced marsh erosion, all of which have been associated with nutrient exposure elsewhere.

[34] (Fig 1). Because Elkhorn Slough contains one of the largest areas of salt marsh in California outside of San Francisco Bay, and its marshes are in such poor condition in terms of habitat loss, both due to marsh drowning and the past history of diking and reclamation to expand grazed lands [24], there is a strong need in the context of this system to know whether nutrient exposure is a primary stressor that needs to be addressed to promote coastal wetland sustainability. In NYC, there are large scale restoration projects underway to rebuild drowning marsh islands, and it would be prudent to better know whether the wastewater effluent and combined sewer overflow discharges play a role in marsh degradation [95].

Results suggest that nutrient pollution was associated with greater rates of organic matter mineralization as measured through carbon dioxide effluxes and litter decomposition, as has been reported by previous studies [22,38]. In addition, our work suggests that coastal wetlands exposed to high levels of water column nutrients also appeared to be in a more decomposed state, as measured through soil humification analysis. While previous work has suggested high nutrient exposure is associated with more sapric soil conditions [17,96], this variable has been difficult to assess, and soil humification analysis may provide some insights into soil quality at other sites. However, nutrient exposure was also associated with greater rates of organic matter production, and consequently, relationships between nutrient exposure and SOM quantity were indirect, and not as simple as has been previously reported [22]. Modeling SOM inventories provided some insight into this observation. While both production and decomposition did scale with nutrient exposure, the disequilibrium between production and decomposition did not because the slopes were different. If nutrient exposure does not affect production and decomposition at the same magnitude, effects of nutrients on SOM, and therefore marsh sustainability, can be difficult to predict.

One major variable that affects SOM inventories is soil texture (Fig 5); thus, we posit that effects of nutrient exposure may be modulated by soil texture as has been suggested by recent work [34]. Soil texture may lead to differential effects of nutrients on plant growth, as finer soils have a greater cation exchange capacity and tend to be more nutrient rich. Finer or organic soils may be more reducing and therefore sulfidic. Lastly, coarser soils may have more oxygen availability, and therefore more rapid rates of SOM turnover. Our study primarily found greater porewater nutrients in finer soils, but other ways we found soil conditions not reflecting soil texture or elevation. For instance, we saw more rapid belowground decomposition in California, even though the soils were finer, and higher porewater sulfide concentrations in New York, even though soils were coarser.

One factor that may lead to enhanced decomposition in nutrient exposed wetlands, that our study did not directly address, is the altered nutrient stoichiometry for organic matter produced under high nutrient availability. Typical C/N/P ratios in terrestrial plants (which include coastal marsh macrophytes) are greater than those preferred by soil microbes. Because plants incorporate greater concentrations of nutrients into their tissues when the nutrients are more bioavailable, altered stoichiometry encourages more rapid and/or more complete mineralization of organic matter [38]. Because of our focus on comparing decomposition rates between marshes, we held the litter sources constant, which might under-estimate organic matter mineralization rates in eutrophic marshes. However, we did observe more rapid rates of belowground decomposition for *Salicornia* than *Spartina* (17%), which had a lower molar CN ratio (36 vs. 41).

With respect to our study of eutrophic marshes, our results suggest that nutrient exposure increases both production and respiration, similar to patterns in other aquatic systems [97].

Because we did not see dramatically lower belowground biomass production under high nutrient exposure, nor dramatically higher rates of decomposition, nor high levels of sulfide concentrations in nutrient exposed marshes associated with these enhanced rates of organic matter cycling (Table 2), this study does not implicate poor water quality as the primary stressor in destabilizing the tidal marshes at our study sites. However, because Elkhorn Slough and Jamaica Bay wetlands are delicately poised at the drowning threshold, and nutrient exposure clearly plays a role in enhancing decomposition, poor water quality cannot be ruled out as a contributing stressor. In NY and Long Island Sound, coastal marsh deterioration has been linked to poor water quality, both because marsh deterioration is more common in western Long Island where coastal areas are more exposed to eutrophic conditions and also at the scale of individual marshes where nutrient loading has been associated with loss of low marsh but sustainability of high marsh [45,54]. However, one factor that complicates the use of gradient studies to elucidate factors contributing to high rates of marsh drowning (5–10% per decade) in NY is the strong covariability between the east-west nutrient gradient and the east-west tidal range gradient [17,45]. Both factors are well known as master variables that control productivity patterns, and so it is difficult to determine which is the main driver.

Because previous studies have reported mixed results about the effect of nutrients on marsh sustainability, our study may provide some clarity to help explain these conflicting reports. Previously, studies have suggested that high nutrient exposure is linked to more sapric soils [8,17,96], enhanced decomposition in association with nitrate exposure [28], and reductions in SOM [22]. Yet, fertilization studies have reported a mix of results, including enhanced belowground biomass, no change, or decreased biomass [40,96,98–101]. Because studies have typically relied on sieving out belowground mass from soil—and it can be nearly impossible to separate live and dead roots—studies that report changes in belowground biomass might often actually be reporting net changes in carbon storage rather than, or in addition to, altered rates of belowground production. Thus, results may be different over the short and long-term, and vary strongly across soil types.

Previous studies have suggested amplified microbial biomass in nutrient enriched marshes may enhance decomposition rates [102], although many more recent studies have focused on community composition and respiration pathways rather than biomass [28,29]. We found consistently higher microbial biomass in marshes exposed to lower nutrients, the opposite of what was expected. However, we did find that soil microbial biomass was strongly correlated with SOM ($r^2$ = 0.83, $p$ = 0.04). Thus, it appears that while addition of wastewater may enhance microbial density or activity in soils [103], these affects are subsumed by the primacy of different soil types. In addition, it has been observed that microbial biomass may change over time in areas exposed to wastewater, with higher biomass in early years and lower biomass in later years [104].

Although not directly related to nutrient exposure, another factor which may play an important role in marsh loss in both Jamaica Bay and Elkhorn Slough is dredging, which has led to increased tidal flooding at both locations [24,47]. Increased tidal flooding may increase exposure to nutrient loading, algal mat accumulation, and amplify soil hydrogen sulfide levels [96]. In some coastal wetlands, increased tidal flooding associated with dredging or subsidence has led increased deposition, and no major negative impacts [105]. However, in Long Island, little sediment is available to build wetland elevation [50]. At Elkhorn Slough, while the watershed is surrounded by agricultural and soil erosion issues are common, water column suspended sediment concentrations are typically quite low. It may be that the low sediment availability, amplified sea level rise, and high nutrient loads—in combination—cause issues with marsh loss not observed when only one stressor is present.

## Regional differences

Our study also highlights functional differences between Atlantic and Pacific wetlands which may be useful for understanding coastal marsh health and integrity. California marshes were found to be more saline, have more fine sediment, different flooding characteristics, and have different ecosystem dominant species. These differences are helpful to report as much coastal wetland literature from the US focuses on the US Atlantic or Gulf Coasts, and comparative studies are rare.

CA marshes were typically hypersaline, which corresponds with the lower precipitation levels and dry summers that characterize the region. They were composed of finer sediments. In CA, soils were 20–22% clay and 70–71% silt, while in NY soils were 6–11% clay and 25–48% silt. These particle size differences are likely linked to differences in geophysical setting. In CA, coastal wetlands are only found in limited heavily sheltered locations due to the active continental margin and heavy surf [106], meaning that apart from river deltas, salt marsh sediment composition is mud-dominated. On Long Island, coastal marshes are typically found in small back-barrier estuaries which receive barrier overwash [107] and consequently coarser particles. With respect to water column nitrate levels, we found the highest values in parts of Elkhorn Slough, where intensive and multi-cropped agriculture dominates nutrient sources, as well as in Sweetwater Marsh, in Morro Bay, where the combination of high density development and a lack of sewer service in the city of Los Osos caused nutrient-enriched shallow groundwater [65]. Although Jamaica Bay is surrounded by high density development, receives wastewater inputs from four wastewater treatment plants, and receives combined sewer overflow and stormwater discharges at more than 100 locations [108], nutrient concentrations in Jamaica Bay were only moderate within the context of this study. This study also highlighted the strong intra-estuary variability in nutrient exposure. Although typically one might think of Jamaica Bay and Elkhorn Slough as eutrophic to hyper-eutrophic, water quality within these estuaries was found to be variable depending on flushing characteristics as well as proximity to nutrient sources or combined sewer overflows. Lastly, the nutrient characteristics of the water column are not necessarily strongly related to nutrient profiles in marshes.

Differences in productivity and decomposition were also apparent between NY and CA marshes. Although standing aboveground biomass may be higher for *Salicornia pacifica*, the west coast dominant, than for *Spartina alterniflora*, the east coast dominant species (values for *S. pacifica* of 2500 g m$^{-2}$ vs. 300–1500 g m$^{-2}$ for *S. alterniflora* [21,109,110], it is clear that *S. pacifica* has lower belowground production, and greater rates of belowground biomass turnover than *S. alterniflora* (Table 2). Depending on soil characteristics, this can drive differences in SOM stocks apparent between east and west coast marshes. For instance, recent studies reported carbon densities four times greater in Northeastern than Californian salt marsh habitats [111,112]. This study found soil carbon inventories that were six times higher in NY than in CA (CA mean = 2500 g m$^{-2}$ vs. NY mean = 16,000 g m$^{-2}$ for the top 20 cm).

Finally, marsh elevations in CA marshes were all above MHW (0.03–0.4m), even though low elevation marsh at Coyote Marsh is on the threshold of drowning and is experiencing erosion. In contrast, NY marsh elevations were typically below MHW (0.37m below MHW to 0.14 m above MHW, with 6 of the 8 sites below MHW). This result is not surprising because CA central coast marshes lack *Spartina foliosa* (the west coast "low marsh" plant), and *Salicornia pacifica* is less tolerant of frequent inundation than cordgrass species. However, this disparity suggests that geomorphology and biogeochemical processes are likely to be distinct in CA and NY marshes as inundation times and patterns are clearly significantly different. As evidence of this, we saw significant porewater nitrate values in CA marshes, whereas in Northeastern marshes porewater nitrate is often unmeasured as most porewater DIN is present in reduced form ($NH_4^+$).

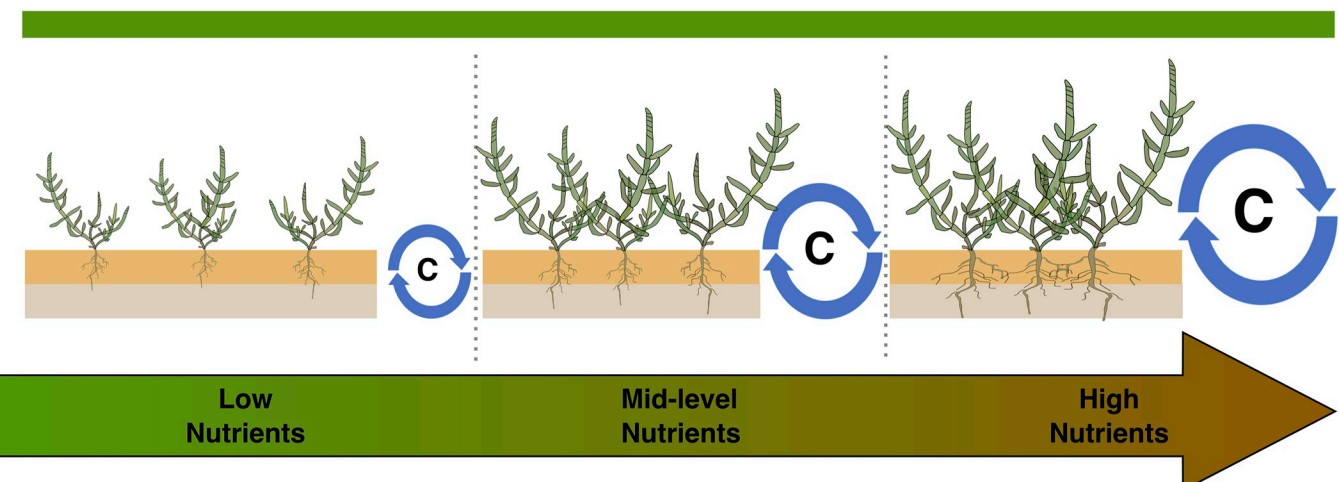

**Fig 6. Relationship between nutrient levels and plant productivity and carbon cycling suggested by this study.**

## Conclusions

In summary, this study found variability in nutrient exposure and its effects. High nutrient exposure was associated with agricultural settings and in watersheds with high density development and treated and untreated wastewater discharges. Low nutrient exposures were associated with watersheds dominated by open space and low-density development (Fig 6). Nutrient exposure enhanced organic matter cycling, which led to both higher rates of production and enhanced decomposition for coastal wetlands, like that described for other aquatic ecosystems. Previous work has suggested that high nutrient exposures can threaten coastal wetland sustainability under climate change, due to enhanced decomposition [8,9,17,32]. While we did find evidence that nutrient exposure enhanced organic matter mineralization, we found that the disequilibrium between organic matter production and decomposition were strongly related to differences in SOM inventories ($r^2$ = 0.61, $p$<0.001). Enhanced nutrient availability supported increased productivity, and increased decomposition, while the difference between these two rates explained variability in SOM. However, because organic matter production and decomposition were not affected at consistent rates, overall impacts to SOM did not scale with nutrient exposure and were influenced by setting. We conclude that while high nutrient loads may have negative effects on CA and NY coastal wetlands, other stressors such as dredging which has enhanced tidal flooding, likely also play important roles. This study also identified important differences between the studied CA and NY wetlands, such as higher salinity, higher elevations, and finer sediments in CA than in NY, that can be used to design comparative investigations, add context to continent-wide assessments, or understand impacts to belowground metrics [113,114].

## Supporting information

**S1 Data. Study data.** Sampling locations, elevations, carbon dioxide efflux, microbial biomass, belowground production, decomposition, humification, stable isotopes, particle size distribution, and porewater sulfides, salinity, nutrients, and pH.
(XLSX)

## Acknowledgments

We acknowledge Kerstin Wasson for helpful insights, Roxanne Johnson, Autumn Oczkowski, and Morgan Schwartz for reviews of early version of this manuscript, and Alana Hanson,

Corey Hamza, Katelyn Szura, Chaniyah Johnson, and John A. Gurak Jr. for field and laboratory assistance. The views expressed in this article are those of the authors and do not necessarily reflect the views or policies of the U.S. Environmental Protection Agency (EPA). Any mention of trade name and products does not imply an endorsement by the U.S. Government or the U.S. EPA. The EPA does not endorse any commercial products, services, or enterprises. This report is Office of Research and Development (ORD) Tracking Number ORD-046457 and it has been reviewed technically by the US EPA's Office of Research and Development, Center for Environmental Measurement and Modeling, Atlantic Coastal Environmental Sciences Division.

## Author Contributions

**Conceptualization:** Elizabeth Burke Watson, Cathleen Wigand.

**Data curation:** Elizabeth Burke Watson, Farzana I. Rahman.

**Formal analysis:** Andrew B. Gray.

**Investigation:** Elizabeth Burke Watson, Farzana I. Rahman, Andrea Woolfolk, Robert Meyer, Nicole Maher, Andrew B. Gray.

**Project administration:** Andrea Woolfolk, Robert Meyer, Nicole Maher, Cathleen Wigand.

**Resources:** Robert Meyer, Nicole Maher.

**Supervision:** Andrew B. Gray.

**Writing – original draft:** Elizabeth Burke Watson.

**Writing – review & editing:** Farzana I. Rahman, Andrea Woolfolk, Nicole Maher, Cathleen Wigand, Andrew B. Gray.

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
