## [Decision Letter · Decision Letter 0]

8 Jun 2022

STANDARD LETTER FROM EDITORIAL OFFICE

PONE-D-22-07419High nutrient loads amplify carbon cycling across California and New York coastal wetlands but with ambiguous effects on marsh integrity and sustainabilityPLOS ONE

Dear Dr. Watson,

Thank you for submitting your manuscript to PLOS ONE. After careful consideration, we feel that it has merit but does not fully meet PLOS ONE’s publication criteria as it currently stands. Therefore, we invite you to submit a revised version of the manuscript that addresses the points raised during the review process.

We look forward to receiving your revised manuscript.

Kind regards,

Just Cebrian

Academic Editor

PLOS ONE

Journal Requirements:

2. Thank you for providing information in the Ethics Statement on the field permits obtained for this study. We ask that you additionally update your Methods section to include this information

4. We note that Figure 2 in your submission contain [map/satellite] images which may be copyrighted. All PLOS content is published under the Creative Commons Attribution License (CC BY 4.0), which means that the manuscript, images, and Supporting Information files will be freely available online, and any third party is permitted to access, download, copy, distribute, and use these materials in any way, even commercially, with proper attribution. For these reasons, we cannot publish previously copyrighted maps or satellite images created using proprietary data, such as Google software (Google Maps, Street View, and Earth). For more information, see our copyright guidelines: http://journals.plos.org/plosone/s/licenses-and-copyright.

Additional Editor Comments:

John C. Stennis Space Center, June 7, 2022

Dear Dr. Watson:

I have received reviews of your paper titled: “High nutrient loads amplify carbon cycling across California and New York coastal wetlands but with ambiguous effects on marsh integrity and sustainability” co-authored with F. I. Rahman, A. Woolfolk, R. Meyer, N. Maher, C. Wigand and A. B. Gray. The Reviewers find merit in the paper, but they also point out a number of clarifications and edits that may help improve it. I have carefully reviewed your document and I find myself in agreement with the Reviewers. The paper describes a nice study and the feedback provided by the Reviewers may help make your paper even better.

With your revised paper, please submit a point-to-point letter explaining how you have addressed each and every one of the comments raised by the Reviewers. Please indicate the lines in the text where those changes have been made. If you disagree and no action is taken, please explain why. That letter will go a long way in helping me reach a final decision. Please make every effort in addressing the Reviewers’ comments satisfactorily since their feedback will certainly benefit your paper.

Thanks for your interest in PLoS ONE and best wishes with your future endeavors

Sincerely

Just Cebrian, PLoS ONE Academic Editor

Reviewers' comments:

Reviewer's Responses to Questions

**Comments to the Author**

1. Is the manuscript technically sound, and do the data support the conclusions?

Reviewer #1: Yes

Reviewer #2: Yes

Reviewer #3: Yes

2. Has the statistical analysis been performed appropriately and rigorously? 

Reviewer #1: Yes

Reviewer #2: Yes

Reviewer #3: Yes

3. Have the authors made all data underlying the findings in their manuscript fully available?

Reviewer #1: Yes

Reviewer #2: Yes

Reviewer #3: Yes

4. Is the manuscript presented in an intelligible fashion and written in standard English?

Reviewer #1: Yes

Reviewer #2: Yes

Reviewer #3: Yes

5. Review Comments to the Author

Reviewer #1: Overall I found this to be a very thorough manuscript that successfully builds the case for the research, outlines an exceptional study, effectively describes the findings in the context of the literature, and presents concluding thoughts objectively. The subject manner, the role of nutrient elevation in marsh sustainability, is incredibly important and timely. The approach of using a series of eight wetlands occurring in Atlantic (NY) and Pacific (CA) settings I found to be a highly innovative approach to enhancing breadth and applicability, and the variability in marsh settings is well detailed. The Methods are thoroughly presented for all but the Data Analysis section, which I did find a bit lacking. If a Generalized Linear Model function was employed then I would expect there to have been at least a brief discussion of the reason underlying the selection of the technique as well as the link function employed. I'm not at all suggesting that the analysis is incorrect, just that a couple of sentences to further explain the reasoning and approach would be beneficial. That said, I thought the use of table 2 to convey findings was effective, and the overall verbal presentation of the results was highly accessible. There were essentially no typographical errors, and I have made just a handful of suggestions to potentially improve the manuscript in the attached PDF, but those are not by any means mandatory and should be only used if the authors see the benefit.

Reviewer #2: This manuscript presents identical physical, chemical, and biological parameters associated with eutrophication, measured in contrasting marsh types on the East and West coasts of the U.S. in order to shed light on the general concept of the role of eutrophication in marsh integrity and sustainability. They do a thorough and exhaustive job of pursuing numerous interactions and explaining in detail the reasoning behind the choice of parameters to measure and how these interactions affect each other and a negative or positive effect on marsh sustainability. The authors' conclusion of ambiguity of the final results sheds light on the difficulty in assigning a definitive effect of eutrophication in all types of marsh under various conditions.

The manuscript is well written and the data is sound. I have attached a tracked changes version of the manuscript with minor typos corrected and grammatical changes. Below I repeat some comments in the margin.

Lines 132-137: These two sentences may explain the main stressors to past, current, and future wetland loss, possibly over-riding eutrophication. These factors, especially the increased tidal range due to dredging, should be repeated at the end of the discussion and in the conclusions as drivers of continued wetland loss, since eutrophication has ambiguous results as the title suggests.

Lines 381-383: Delete this sentence from the figure caption. It can be inserted into the text of the manuscript, perhaps as the last sentence in the above paragraph (line 378) or the last sentence in this section (line 395).

Lines 597-600: As with the Figure 4 caption, these 2 sentences are better moved to the text in the conclusions, perhaps at line 581 after “(Fig 6).”, or elsewhere in the paragraph.

Reviewer #3: General comments:

The Authors have a well written manuscript that outlines a study that measured a suite of

belowground characteristics across salt marshes of varied nutrient exposure. The Authors

noted salt marsh “drowning” in areas of high nutrient exposure, which inspired this study

to determine how high nutrient exposure may influence belowground characteristics that

would ultimately lead to marsh loss. The Authors sampled two sites within four estuaries,

two within New York and two within California, for a total of eight salt marsh sites that

ranged a nutrient exposure gradient. The Authors measured factors such as litter

breakdown, belowground productivity via in-growth bags, soil respiration and nutrient

content, and soil microbial biomass. The Authors found sites with higher nutrient

exposure had higher belowground productivity as well as faster organic matter

breakdown. Due to these results, the Authors concluded that high nutrient exposure alone

could not be the cause of marsh drowning, but certainly could be a confounding factor

due to the increased mineralization.

The Authors present a nicely put together manuscript. The study design, by incorporating

estuaries of various nutrient exposure, cleverly uses the natural landscape to elucidate

large trends to aid in our understanding of marsh stability. The data analyses and

accompanying tables and figures are comprehensive and easily discernible. The writing is

clean, clear, and concise.

There are a few items that should be addressed and/or expanded upon in the Discussion.

In Line 472, there is mention that soil texture can influence nutrient impacts. It would be

good to not only summarize the previous work that is referenced on how soil texture can

influence nutrient impacts but also apply this information to the study results. The

California and New York sites varied in soil texture (e.g., New York had much higher

sand content) and since California and New York sites are not evenly distributed along

the nutrient exposure gradient (e.g., California sites are the two highest nutrient exposure

sites) the soil texture differences may have played a part in findings beyond just the

nutrient exposure gradient. The Discussion would also benefit on some discussion of the

differences between the two dominant plant types used in this study. In the Results, there

is mention of Salicornia pacifica “woody nature” (Line 375-377), but this should be

mentioned again in the Discussion, along with the differences in breakdown rates

between succulent forb and graminoid leaves.

Specific comments:

Line 65-70: Consider adding more salt marsh related examples.

Line 82: Appears two spaces after the comma.

Line 85-86: Sentence is a bit awkwardly worded; consider revising.

Line 91-93: Briefly articulate how “decomposition is problematic…to wetland integrity.”

Line 96: Consider revising or including an example of what is meant in terms of “soil

integrity.”

Line 102-105: Revise and make more concise.

Line 141-142: Suggest including the other two estuaries studied.

Line 145: Define what soil integrity means in the context of this study, especially as to

differentiate from earlier mentioned marsh integrity, which soil integrity appears to one

of its measures.

Line 166: What is exactly meant by “nutrient availability conditions”? Is this related to

the “nutrient exposure” that is stated later?

Line 169-172: Would be nice to re-state which two are the eutrophic estuaries. Although

it would appear that really one site in each of the two CA estuaries are eutrophic and two

are not. This may need to be stated as from the Introduction, it would seem that the entire

Elkhorn Slough estuary is considered eutrophic. Additionally, are there any signs of

marsh drowning in the other two estuaries stated here, like the two mentioned in the

Introduction?

Line 200: Data were not available

Line 231: State that live and dead material were not separated (information inferred from

Line 510-511).

Line 238: How was material dried prior to deployment?

Line 245: What were the intervals that the bag collections occurred?

Line 260: Soil surface temperatures?

Line 311-313: Consider sentence revision as reads a bit confusing.

Line 322-325: Consider sentence revision.

Line 455: “Here” isn’t needed.

Line 466-469: Consider sentence revisions as reads a bit confusing.

Line 475-483: This paragraph acknowledges an important aspect and is appreciated.

Perhaps here too can add the differences in nutrient ratios between succulent and

graminoid leaves.

Line 518-522: This should be expanded a bit more and as just stating “substrate plays an

extremely important role” doesn’t seem to fully flesh out why the results were not as

expected.

Line 591-594: Suggest also including how these regional differences may help to

understand impacts to belowground metrics.

Figure 4: It is unclear why New York and California sites are separated here, but the rest

of the analyses compare all sites together along nutrient exposure gradient. In addition,

the colors are a bit hard to see.

6. PLOS authors have the option to publish the peer review history of their article (what does this mean?). If published, this will include your full peer review and any attached files.

Reviewer #1: No

Reviewer #2: No

Reviewer #3: No

---

## [Author Response · Author response to Decision Letter 0]

2 Aug 2022

Please see the uploaded file, "Response document," which includes a fully formatted response to all editorial and reviewer comments and suggestions.

---

## [Editor Report · Decision Letter 1]

5 Aug 2022

High nutrient loads amplify carbon cycling across California and New York coastal wetlands but with ambiguous effects on marsh integrity and sustainability

PONE-D-22-07419R1

Dear Dr. Watson,

We’re pleased to inform you that your manuscript has been judged scientifically suitable for publication and will be formally accepted for publication once it meets all outstanding technical requirements.

Kind regards,

Just Cebrian

Academic Editor

PLOS ONE
---

## [Editor Report · Acceptance letter]

31 Aug 2022

PONE-D-22-07419R1 

High nutrient loads amplify carbon cycling across California and New York coastal wetlands but with ambiguous effects on marsh integrity and sustainability 

Dear Dr. Watson:

I'm pleased to inform you that your manuscript has been deemed suitable for publication in PLOS ONE. Congratulations! Your manuscript is now with our production department. 

Kind regards, 

on behalf of

Dr. Just Cebrian 

Academic Editor

PLOS ONE